# Fast and Complete Destruction of the Anti-Cancer Drug Cytarabine from Water by Electrocatalytic Oxidation Using Electro-Fenton Process

**Sule Camcioglu [1,2], Baran Özyurt [1,2], Nihal Oturan [2], Clément Trellu [2] and Mehmet A. Oturan [2,*]**

[1] Department of Chemical Engineering, Faculty of Engineering, Ankara University, Tandogan, Ankara 06100, Turkey

[2] Laboratoire Géomatériaux et Environnement EA 4508, Université Gustave Eiffel, CEDEX 2, 77454 Marne-la-Vallée, France

\* Correspondence: mehmet.oturan@univ-eiffel.fr

**Abstract:** The fast and complete removal of the anti-cancer drug cytarabine (CYT) from water was studied, for the first time, by the electro-Fenton process using a BDD anode and carbon felt cathode. A catalytic amount ($10^{-4}$ M) of ferrous iron was initially added to the solution as catalyst and it was electrochemically regenerated in the process. Complete degradation of 0.1 mM (24.3 mg L$^{-1}$) CYT was achieved quickly in 15 min at 300 mA constant current electrolysis by hydroxyl radicals ($^{\bullet}$OH) electrocatalytically generated in the system. Almost complete mineralization (91.14% TOC removal) of the solution was obtained after 4 h of treatment. The mineralization current efficiency (MCE) and energy consumption (EC) during the mineralization process were evaluated. The absolute (second order) rate constant for the hydroxylation reaction of CYT by hydroxyl radicals was assessed by applying the competition kinetics method and found to be $5.35 \times 10^9$ M$^{-1}$ s$^{-1}$. The formation and evolution of oxidation reaction intermediates, short-chain carboxylic acids and inorganic ions were identified by gas chromatography-mass spectrometry, high performance liquid chromatography and ion chromatography analyses, respectively. Based on the identified intermediate and end-products, a plausible mineralization pathway for the oxidation of CYT by hydroxyl radicals is proposed.

**Keywords:** advanced oxidation processes (AOPs); electrochemical AOPs; electro-Fenton; cytarabine; wastewater treatment

## 1. Introduction

Anti-cancer drugs are only partially metabolized and then discharged into the sewage system from hospitals and houses with the urine and/or feces of patients, in the form of active substance and metabolites [1]. Cytarabine (CYT) is an anti-cancer drug which belongs to the category of antimetabolites [2]. It is an analogue of deoxycytidine and is used to inhibit the growth of unregulated cancer cells by interfering with the DNA synthesis of the cell [3–5]. The urinary excretion rate for CYT is approximately 10% and its half-life degradation in the human body is 1–3 h [6]. CYT has been detected in concentrations up to 9.2, 14, 13 ng L$^{-1}$ in the influents and effluents of sewage treatment plants and surface waters, respectively [7]. These drugs cannot be successfully removed from water by conventional treatment methods due to their resistance to physical and biological treatments [8,9]. Therefore, effluents from wastewater treatment plants are considered to be the main source of the release of anti-cancer drugs and their metabolites into the aquatic environment [10]. Chronic exposure to parent anti-cancer drugs and their transformation products is considered to be harmful to aquatic organisms and humans due to their highly potent mechanism of action [10]. Therefore, it is of great importance to develop effective treatment methods to prevent the release of these drugs and their degradation intermediates into the natural water media [11].

On the other hand, advanced oxidation processes (AOPs) are becoming more and more important technologies for the efficient removal of non-biodegradable, persistent and toxic pollutants from water [12–15]. The main advantages of these processes are that they are relatively easy to operate due to the use of safe and environmentally friendly reagents, the absence of mass transfer restrictions and the short reaction time [16,17]. AOPs are based on the principle of in situ generation of strong oxidants, hydroxyl radicals ($^\bullet$OH) that can oxidize all types of organic pollutants in water [18,19]. Various AOP mechanisms have been proposed, and these mechanisms are classified depending on how the hydroxyl radical formation occurs: by chemical, photochemical, sonochemical or electrochemical AOPs [12,20]. The oldest and most used AOP is the chemical Fenton process which is based on the use of a mixture of $H_2O_2$ and ferrous iron to form hydroxyl radicals through the Fenton reaction (Equation (1)) [12].

$$H_2O_2 + Fe^{2+} \rightarrow Fe^{3+} + OH^- + {}^\bullet OH \tag{1}$$

This process presents some drawbacks such as the external addition of reactants, the risks related to their transport and storage, and the involvement of several wasting reactions reducing process efficiency [21]. One of the ways to improve the Fenton process efficiency is its combination with electrochemical technology [22]. Among the AOPs, electrochemical advanced oxidation processes (EAOPs) are promising in terms of industrial applicability with their advantages including low operating cost [23,24] and high removal efficiency of pollutants [25–27]. Electro-Fenton is one of the most popular EAOPs for the removal of toxic and/or bio-refractory pollutants [28–30]. In this process, $H_2O_2$ is continuously electrogenerated at a suitable cathode by the electro-reduction of $O_2$ through the reaction given in Equation (2) [31,32]. $H_2O_2$ reacts with externally added catalyst ($Fe^{2+}$) to generate homogeneous hydroxyl radicals from Fenton's reaction (Equation (1)) at an optimal pH of around 3 [23,25]. Then, the catalyst $Fe^{2+}$ is continuously electro-regenerated from the reduction of $Fe^{3+}$ formed through the Fenton reaction according to the reaction given in Equation (3) [33]. The continuous electrogeneration of $H_2O_2$ and electro-regeneration of $Fe^{2+}$ create a catalytic cycle accelerating the oxidation/mineralization processes [34].

$$O_2 + 2H^+ + 2e^- \rightarrow H_2O_2 \tag{2}$$

$$Fe^{3+} + e^- \rightarrow Fe^{2+} \tag{3}$$

The use of an anode material (M) with a high $O_2$ evolution overpotential such as a BDD electrode in the electro-Fenton process allows the production of hydroxyl radicals at the anode surface by water oxidation (Equation (4)) [35,36]. These heterogeneously formed radicals, noted M($^\bullet$OH), are barely adsorbed on the electrode surface and behave as quasi-free radicals in the vicinity of the anode [37]. The generation of these supplementary hydroxyl radicals in the system enhances the efficiency of the process [38,39].

$$M + H_2O \rightarrow M({}^\bullet OH) + H^+ + e^- \tag{4}$$

The electro-Fenton process is a promising method in terms of industrial applicability owing to its high pollutant removal efficiency and simple operating conditions [40]. Compared to the classical Fenton process, the electro-Fenton process needs a significantly smaller amount of ferrous ions (as catalyst) [41,42]. In addition, $H_2O_2$ is in situ generated in the process [43]. The use of the appropriate anode and cathode affects the efficiency of the process [44–46].

In this study, we investigated the kinetics of oxidation and mineralization of CYT solutions using the electro-Fenton process with a BDD anode and carbon felt cathode. To the best of our knowledge, this constitutes the first work on the efficient removal of CYT from contaminated water. We found only one report [3] treating anodic oxidation of CYT in which lower oxidation kinetics and mineralization rates were reported. The present study reports fast oxidative degradation kinetics (complete degradation in 15 min) and

almost complete mineralization (91.14% TOC removal after a 4 h treatment). Furthermore, the reaction intermediates formed during the oxidation process, the short-chain carboxylic acids as final by-products before complete mineralization and mineral end-products were identified using HPLC, GC-MS and ion chromatography analysis methods. These data allow us to propose a plausible mineralization pathway of CYT by hydroxyl radicals generated in the electro-Fenton process.

## 2. Results and Discussion

### 2.1. Oxidative Degradation Kinetics of CYT

The applied current is the key parameter controlling the regulation of the amounts of generated hydroxyl radicals through Equations (1)–(4) and therefore affecting the efficiency of the electro-Fenton process [47,48]. Figure 1a depicts the decay kinetics of the CYT concentration when using BDD/carbon felt cells to elucidate the effect of current on degradation kinetics. As can be seen in Figure 1a, the complete degradation of 0.1 mM CYT was quickly achieved in all cases, verifying that the drug was efficiently oxidized by hydroxyl radicals both in the bulk solution and at the surface of the anode. Therefore, a second order reaction kinetics of CYT can be written as follows:

$$CYT + {}^{\bullet}OH \rightarrow products \tag{5}$$

$$\frac{d[CYT]}{dt} = -k[CYT][{}^{\bullet}OH] = -k_{app}[CYT] \tag{6}$$

where $k_{app}$ is the apparent rate constant. Since the electrolytic cell is operated under galvanostatic conditions, hydroxyl radicals are produced at a constant rate in the process and it is convenient to assume that they are consumed as soon as they are produced. Thus, under these conditions, the second order reaction becomes a pseudo-first-order reaction. From the integration of Equation (6), Equation (7) is obtained as follows.

$$\ln \frac{[CYT]_0}{[CYT]_t} = k_{app} \times t \tag{7}$$

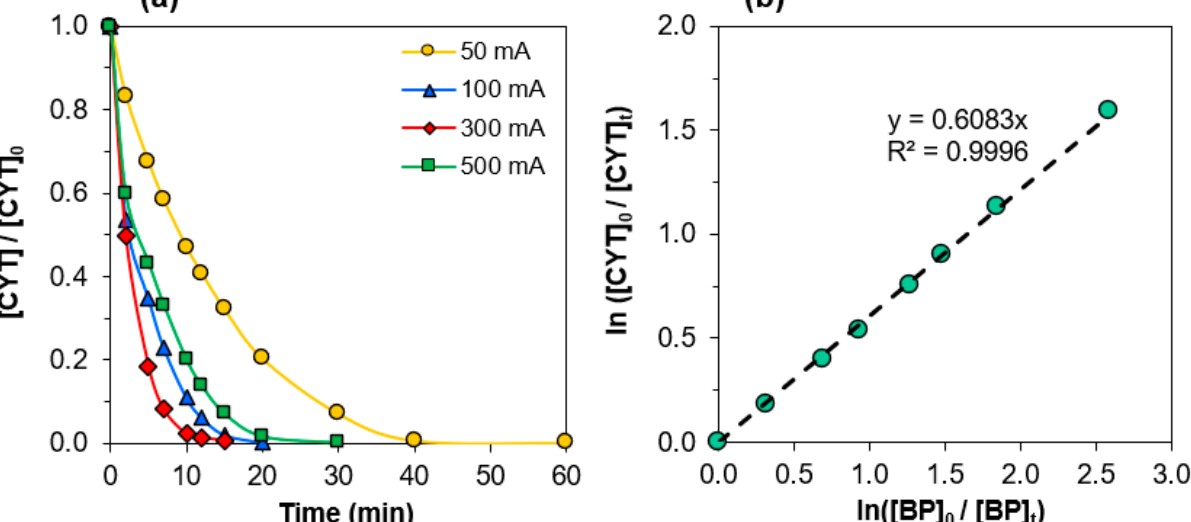

**Figure 1.** Effect of current on CYT concentration decay (**a**) and determination of absolute rate constant ($k_{CYT}$) of the reaction between CYT and hydroxyl radicals at 50 mA (**b**). Experimental conditions: pH = 3, [Na$_2$SO$_4$] = 50 mM, [Fe$^{2+}$] = 0.1 mM.

The slope of ln([CYT]$_0$/[CYT]$_t$) versus t plot gives the $k_{app}$ values [49] and results obtained are given in Table 1 with coefficients of determination (R$^2$) for CYT degradation. According to the high values of R$^2$ obtained in the experiments, it was concluded that the

pseudo-first-order kinetics model for oxidation of CYT by hydroxyl radicals generated in the electro-Fenton process was described successfully.

**Table 1.** Apparent rate constants for cytarabine (CYT) degradation by electro-Fenton.

| I (mA) | $k_{app}$ (min$^{-1}$) | $R^2$ |
|--------|------------------------|-------|
| 50 | 0.0825 | 0.9893 |
| 100 | 0.2367 | 0.9819 |
| 300 | 0.3616 | 0.9966 |
| 500 | 0.1688 | 0.9880 |

The decay of CYT concentration was improved by the increase in current value up to 300 mA for which complete degradation of CYT was achieved in 15 min. Further increase in current value did not enhance the degradation efficiency. The decrease in the oxidative degradation kinetics at high current values may be related to the enhancement of the rate of the side reactions that inhibit the formation or promote the consumption of hydroxyl radicals. The most important of these side reactions are (I) evolution of $H_2$ at the cathode which competes with the formation of $H_2O_2$ (Equation (8)) and (II) the evolution of $O_2$ in the bulk (Equation (9)) and on the anode surface (Equation (10)) which slows down the oxidation rate [23,25].

$$2H^+ + 2e^- \rightarrow H_2 \tag{8}$$

$$2H_2O \rightarrow O_2 + 4H^+ + 4e^- \tag{9}$$

$$2M(^\bullet OH) \rightarrow 2M + O_2 + 2H^+ + 2e^- \tag{10}$$

The absolute rate constant for the hydroxylation reaction of CYT by hydroxyl radicals was assessed by applying the competition kinetics method. This method is based on the competitive degradation of the target molecule and a standard competitor with a well-known rate constant [50]. Thus, benzophenone (BP) with an absolute (second order) rate constant of $k_{BP}$ of $8.8 \times 10^9$ M$^{-1}$s$^{-1}$ was selected as the standard competitor [51]. The absolute rate constant was then determined according to the Equation (11):

$$\ln\left(\frac{[CYT]_0}{[CYT]_t}\right) = \frac{k_{CYT}}{k_{BP}} \times \ln\left(\frac{[BP]_0}{[BP]_t}\right) \tag{11}$$

Here, $k_{CYT}$ and $k_{BP}$ are the absolute rate constants for the oxidation reaction by hydroxyl radicals for CYT and benzophenone, respectively. In order to avoid the interference of the oxidation products, experiments were performed at 50 mA and on a short electrolysis time. A solution containing the same concentration (0.1 mM) of both CYT and benzophenone was oxidatively degraded in the presence of 0.1 mM Fe$^{2+}$ at pH 3, and $k_{CYT}$ was then calculated as $5.35 \times 10^9$ M$^{-1}$ s$^{-1}$ from the slope of the line given in Figure 1b.

### 2.2. Mineralization of CYT Aqueous Solution

The complete oxidative degradation of the target pollutant does not mean that the solution to be treated is cleaned effectively due to the generation of intermediate products which can be more toxic or refractory compared to mother molecule [52]. For an effective treatment, almost total removal of mother pollutants and their oxidation intermediates is required. Hence, the mineralization of 0.1 mM CYT solution was performed at different current values with regard to TOC removal efficiency. As can be seen from Figure 2a, the TOC removal rate increased with the increments of current from 50 to 500 mA, as expected from the enhanced production of hydroxyl radicals. This fact can be explained by the high oxygen evolution overpotential of the BDD anode allowing the formation of high amounts of heterogeneously formed hydroxyl radicals (Equation (4)) as well as homogeneously formed hydroxyl radicals in the bulk, due to the increased rate of

electrogenerated $H_2O_2$ and fast electro-regeneration of $Fe^{2+}$ promoting Fenton's reaction (Equations (1)–(3)) [53,54]. According to Figure 2a, the percentage of TOC removal after a 4 h treatment was observed as 91.14 and 94.77% at 300 mA and 500 mA, respectively, showing that almost complete mineralization had been achieved. When increasing the applied current value, the difference in the TOC decay rate observed at lower current values was greater than that observed between higher currents. This behavior can be explained by the acceleration of parasitic non-oxidative reactions of the hydroxyl radicals at higher current values, mainly heterogeneously formed hydroxyl radical oxidation to $O_2$ (Equation (10)) and the recombination of homogeneously formed hydroxyl radicals to $H_2O_2$ (Equations (12) and (13)). It was also observed that TOC showed a very sharp decay during the first two hours of electrolysis followed by a much slower removal rate. This phenomenon can be explained by the loss of organic matter in the solution on the one hand, and the formation of more recalcitrant organic compounds such as short–chain carboxylic acids, on the other hand [54–56].

$$2M(^\bullet OH) \rightarrow 2M + H_2O_2 \tag{12}$$

$$^\bullet OH + {^\bullet OH} \rightarrow H_2O_2 \tag{13}$$

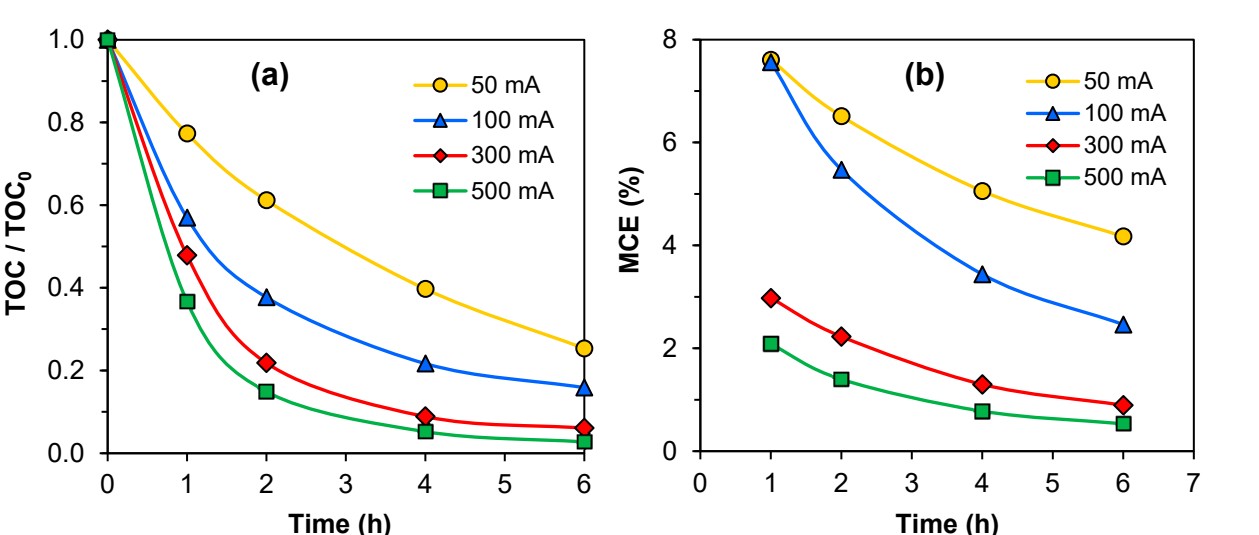

**Figure 2.** Effect of current on normalized TOC removal of 0.1 mM CYT in 50 mM $Na_2SO_4$ solution containing 0.1 mM $Fe^{2+}$ (**a**) and evolution of MCE over time (**b**). Experimental conditions: pH = 3 in, $[Na_2SO_4]$ = 50 mM, $[Fe^{2+}]$ = 0.1 mM.

Figure 2b represents the mineralization current efficiency (MCE) values that reflect the proportion of the current used for mineralization of CYT. The MCE curves are getting closer with the increase in current and in TOC decay. It is observed that MCE progressively decays after showing a maximum value at the beginning of electrolysis, suggesting the loss of organic matter and the generation of persistent by products (such as short-chain carboxylic acids) occurring just after the rapid oxidation of organics under the attack of hydroxyl radicals. After 6 h of electrolysis, MCE values were 4.17, 2.46, 0.89 and 0.53% for 50, 100, 300 and 500 mA current intensities, respectively. This can be explained by the enhancement of the rate of waste reactions that consume hydroxyl radicals or electrical energy spent for the generation of less powerful oxidation species in side reactions, such as $H_2$ evolution at the cathode and $O_2$ evolution at the anode [57], as detailed above.

The MCE is an important parameter in view of the large application of a process. In general, the depletion of MCE has been reported to be associated with an increase in the energy consumption (EC) per TOC unit removed [17]. The evolution of EC for current values ranging from 50 to 500 mA is shown in Figure 3 to evaluate the cost effectiveness of the process. According to the results, higher current and longer electrolysis time result in

high EC. The increase in allied current enhances the mineralization of organic pollutants up to a maximum value followed by a decrease due to the acceleration of non-oxidizable parasitic reactions and the generation of recalcitrant intermediates, as discussed above.

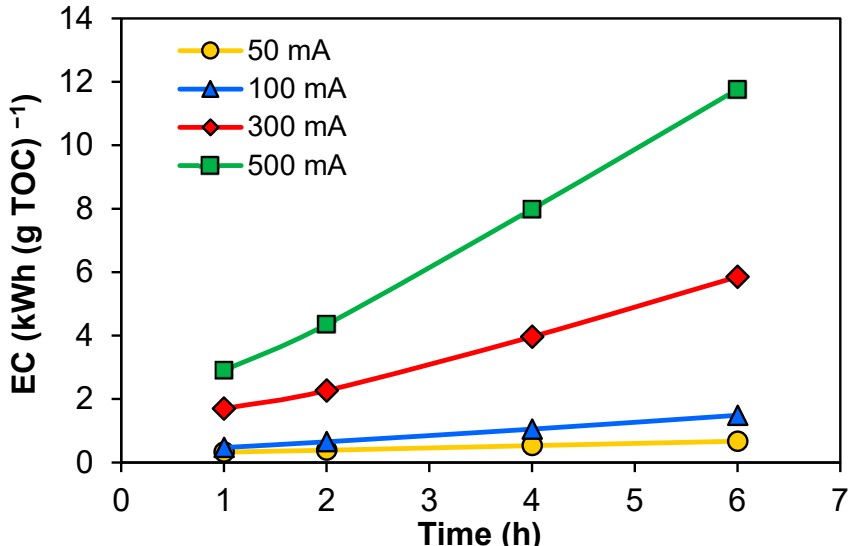

**Figure 3.** Evolution of EC during the electrolysis of 0.1 mM CYT in 50 mM of $Na_2SO_4$ solution containing 0.1 mM $Fe^{2+}$ as a function of current.

### 2.3. Formation and Evolution of Carboxylic Acids and Inorganic Ions

The oxidative degradation of organic pollutants by hydroxyl radicals leads to the formation of a variety of short-chain carboxylic acids, which are the lowest molecular weight organic species and ultimate reaction intermediates before the total mineralization step consisting of transformation to $CO_2$, $H_2O$ and inorganic ions during AOPs [23,54]. These species are reported to be less reactive against the attack of hydroxyl radicals and generally need longer electrolysis times for complete mineralization [58,59]. The identification and evolution of carboxylic acids during the mineralization of CYT in the electro-Fenton process are shown in Figure 4a. Ion-exclusion chromatograms allowed the detection of four carboxylic acids with well-defined peaks corresponding to oxalic, oxamic, pyruvic and formic acids at retention times ($t_R$) of 6.64, 9.87, 11.28, 13.55 min, respectively. The concentration profile of these carboxylic acids followed a quick accumulation trend at the early stage of the electrochemical treatment with regard to the quick oxidation of the target molecule and its organic cyclic intermediates by electrogenerated hydroxyl radicals [60]. However, the concentration of these species gradually decreased with the electrolysis time since they were present in the medium mainly as Fe(III)-carboxylate complexes which require longer destruction times due to their lower reactivity with hydroxyl radicals [49,53]. As seen from Figure 4a, pyruvic acid was detected at trace level and disappeared after about 30 min electrolysis. Oxalic acid which is regarded as the ultimate end product before total mineralization [61,62] was obtained at maximum concentration (0.28 mM) at 60 min. In addition, formic and oxamic acids reached maximum values at 10 and 60 min, respectively. It should be noted that these two carboxylic acids were completely removed after 90 and 330 min of electrolysis, respectively, whereas the concentration of oxalic acid slowly decreased to a low amount (0.023 mM) during electrolysis with the BDD anode. This is consistent with low TOC values under the same experimental conditions, as presented in Section 3.2.

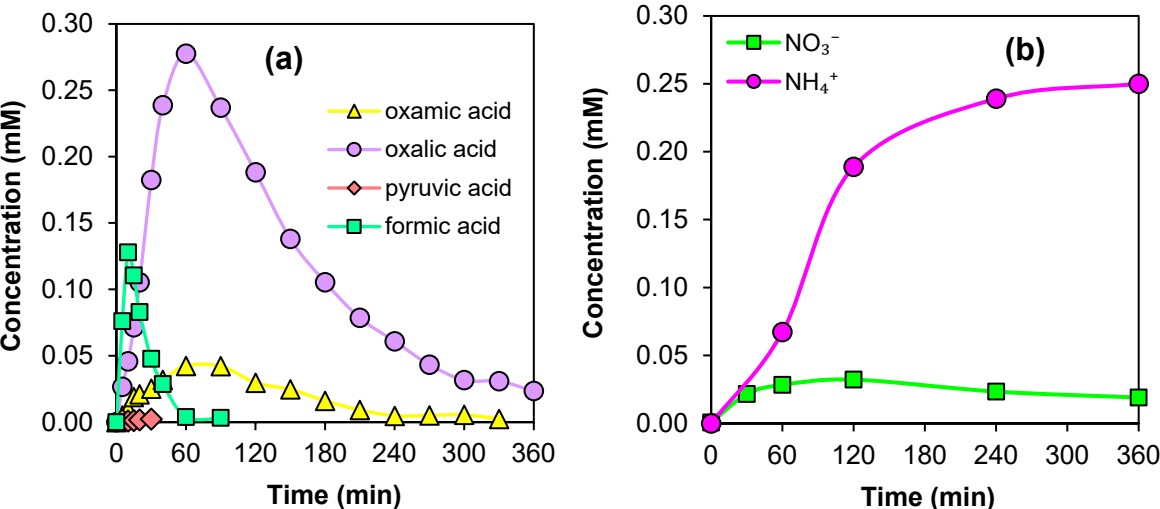

**Figure 4.** Evolution of short-chain carboxylic acids (**a**) and inorganic ions (**b**) during electrolysis of 0.1 mM CYT solution containing 0.1 mM $Fe^{2+}$ (as catalyst) during electro-Fenton treatment with BDD anode and carbon-felt cathode at pH = 3. Applied current was 100 mA for carboxylic acids whereas it was 300 mA for inorganic ions. The concentration of the supporting electrolyte ($Na_2SO_4$) was 50 mM for carboxylic acids while it was 15 mM for inorganic ions to avoid the interference of $Na^+$ ions on analysis.

The heteroatoms in the structure of organic molecules are oxidized to inorganic ions that are released in the solution during mineralization. The identification and evolution of inorganic ions (corresponding to heteroatoms in the structure of mother organic pollutant) represent another important indicator for the mineralization of organic pollutants [54,56]. CYT contains three N atoms in its initial structure which were expected to be mineralized into inorganic ions $NO_3^-$ and $NH_4^+$ upon cleavage of the covalent bonds in the molecule. The inorganic ions formed during the mineralization of 0.1 mM CYT solution were identified and quantified by ion chromatography. The evolution of $NO_3^-$ and $NH_4^+$ is depicted in Figure 4b, while $NO_2^-$ ions were not detected in the solution. It can be seen that the concentration of $NH_4^+$ ions which continuously accumulated throughout the electrolysis time was significantly higher than that of $NO_3^-$ in the solution. At the early stages of the electrolysis, the concentration of $NH_4^+$ increased with treatment time and gradually reached a plateau. The concentration of $NO_3^-$ remained almost constant after 1 h electrolysis and could be related to its reduction to $NH_4^+$ ion on the carbon felt cathode in agreement with previous reports [58,63]. After 6 h of electrolysis, the concentration of $NH_4^+$ ions in the final solution was determined as 0.250 mM (83.3% of initial N) whereas the concentration of $NO_3^-$ was 0.019 mM (6.3% of initial N). The sum of these two inorganic ions released into the solution was calculated as 0.27 mM representing 90% of the total nitrogen content in the initial solution. The slight loss of N could be explained with the generation of gaseous nitrogen compounds ($N_2$, $N_2O_5$, $NO_x$) via the cathodic reduction of $NO_3^-$ [64].

### 2.4. Mineralization Pathway for CYT

The identification of the aromatic oxidation products formed during the electro-Fenton treatment was performed using GC-MS analysis (Table 2). These data, combined with those related to carboxylic acids and inorganic end-products (detailed in Section 2.3), allowed us to propose a plausible mineralization pathway of CYT by hydroxyl radicals generated during the electro-Fenton process (Table 2, Figure 5). The intermediates I to IV were identified thanks to the fragmentation analysis of the GC-MS spectrum while the intermediates V to VIII were identified by ion-exclusion HPLC analysis. The proposed pathway starts with the attack of hydroxyl radicals on CYT, leading to oxidative cleavage into molecules I (D-ribofuranose) and II (cytosine). Compounds III and IV were the result of the hydroxylation of cytosine. Further attacks of hydroxyl radicals cause oxidative ring-opening reactions

of these aromatic structure leading to the formation of carboxylic acids (compounds V to VIII) and inorganic ions through deamination and decarboxylation mechanisms. The mineralization of the carboxylic acids to $CO_2$, water and inorganic ions constitutes the last stage of mineralization in agreement with TOC and inorganic ions analysis.

**Table 2.** Intermediate products identified using GC-MS and HPLC during the mineralization of CYT by electro-Fenton process with BDD anode.

| Compound | Molecular Mass (g mol$^{-1}$) | Retention Time (min) | Molecular Structure | Analytical Technique |
|---|---|---|---|---|
| I | 150 | 13.46 | | GC-MS |
| II | 111 | 20.07 | | GC-MS |
| III | 113 | 23.58 | | GC-MS |
| IV | 129 | 19.96 | | GC-MS |
| V | 90 | 6.64 | | HPLC |
| VI | 46 | 13.55 | | HPLC |
| VII | 89 | 9.87 | | HPLC |
| VIII | 88 | 11.28 | | HPLC |

**Figure 5.** Proposed reaction pathway for mineralization of CYT by hydroxyl radicals based on TOC data and identified oxidation intermediates, short-chain carboxylic acids and mineral end-products.

## 3. Material and Methods

### 3.1. Chemicals

CYT, $C_9H_{13}N_3O_5$, with purity >98%, was purchased from Acros Organics and was used without further purification. Methanol ($CH_3OH$) and phosphoric acid ($H_3PO_4$) used in the preparation of HPLC eluents were supplied by Sigma-Aldrich and Fluka, respectively. Sodium sulfate ($Na_2SO_4 \geq 99\%$ purity) used as supporting electrolyte and iron (II) sulphate heptahydrate ($FeSO_4 \cdot 7H_2O \geq 99.5\%$ purity) used as source of the catalyst were provided by Merck and Sigma-Aldrich, respectively. Sulfuric acid ($H_2SO_4$) and sodium hydroxide (NaOH) used to adjust solution pH were of analytical grade from Acros Organics and Fluka, respectively. HPLC eluents and CYT working solutions were prepared with ultra-pure water obtained from a Millipore Milli-Q system with resistivity > 18 M$\Omega$ cm at 25 °C. Organic solvents and other chemicals used were either HPLC or analytic grade from Sigma-Aldrich. Oxalic ($C_2H_2O_4$), oxamic ($C_2H_3NO_3$), pyruvic ($C_3H_4O_3$) and formic ($CH_2O_2$) acids used as standards for quantifying short-chain aliphatic carboxylic acids generated during electro-Fenton treatment were obtained from Acros Organics, Fluka and Alfa Aesar.

### 3.2. Electrolytic System

Homogeneous electro-Fenton experiments were performed in a 250 mL undivided cylindrical glass cell. To homogenize the solution, a magnetic stirrer and compressed air bubbling were used. The bubbling was started 5 min before the beginning of the experiment to provide saturated oxygen conditions in the solution for $H_2O_2$ generation. BDD film on niobium support (CONDIAS GmbH, Itzehoe, Germany) was used as anode and unmodified 3D carbon felt electrode (MERSEN, Paris, France) was used as cathode. Both electrodes had a surface area of $4 \times 8$ cm$^2$. Experiments were carried out at room temperature and at pH 3. This value is well established to be optimal for Fenton process and all related processes including electro-Fenton process [12,23,30]. The pH was measured with a CyberScan pH 1500 pH-meter from Eutech Instruments. The electrochemical cell was powered by a Hameg HM8040 triple DC power supply. A schematic illustration of the electrochemical experimental system is presented in Figure 6. Experiments were performed under constant current electrolysis conditions ranging from 50 mA to 500 mA on a 220 mL aqueous solution containing 0.1 mM CYT as target pollutant, 0.1 mM $Fe^{2+}$ as catalyst. Electrolyses were performed in 50 mM $Na_2SO_4$ as supporting electrolyte since it behaves inert under electro-Fenton conditions. 50 mM is sufficient to provide conductivity for current flow without loss of electrical energy [12,23]. Samples were taken during electrolysis at pre-set time intervals in order to evaluate the concentration for decay kinetics of CYT, mineralization rate, carboxylic acid evolution and inorganic ions released. All experiments for degradation kinetics and TOC abatement were performed at least twice and the mean values are reported in the figures.

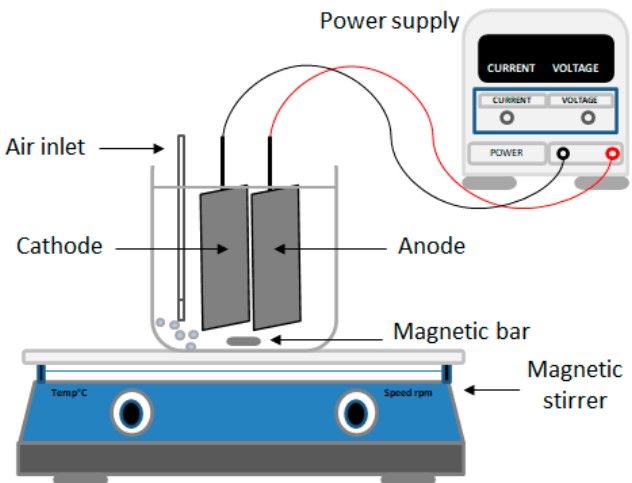

**Figure 6.** Schematic illustration of electrolytic system used in this work.

### 3.3. Analytical Procedures

The CYT concentration was measured by HPLC using a Merck Lachrom Liquid chromatography system equipped with a quaternary pump L-7100, fitted with a Purospher RP 18, 5 µm, 25 cm × 4.6 mm (i.d.) column from VWR International (Fontenay-sous-Bois, France) at 40 °C and coupled with a L-7455 diode array detector from Hitachi (Fontenay-sous-Bois, France). The analysis of CYT was carried out with a mobile phase composed of methanol/water (with 0.4% phosphoric acid) 2:98 (*v/v*) mixture in isocratic elution mode at a flow rate of 0.3 mL min$^{-1}$. The detection was performed at the wavelength of 271 nm. Samples of 20 µL were injected into the system and measurements were controlled through EZ-Chrom Elite (Agilent, Palo Alto, CA, USA) 3.1 software.

The extent of mineralization during the treatment of CYT was monitored from the abatement of its total organic carbon (TOC) value using a Shimadzu TOC-V$_{CSH}$ analyzer according to the thermal catalytic oxidation principle. The carrier gas was oxygen with a flow rate of 150 mL min$^{-1}$. The temperature in the oven was 680 °C. Platinum was

used as catalyst in order to carry out the total combustion reaction at this temperature. Calibration of the analyzer was carried out with potassium hydrogen phthalate standard. Reproducible TOC values were obtained using the non-purgeable organic carbon method with an accuracy of $\pm 2\%$ by injecting 50 µL aliquots into the analyzer.

TOC values were also utilized for determination of mineralization current efficiency (MCE) and energy consumption (EC). MCE was calculated according to the following equation (Equation (14)) [14]:

$$\text{MCE}(\%) = \frac{n\,F\,V_S\,\Delta(\text{TOC})_{\text{exp}}}{4.32 \times 10^7\,m\,I\,t} \times 100 \tag{14}$$

Here, F is the Faraday constant (96485 C mol$^{-1}$), Vs is the solution volume (L), $\Delta(\text{TOC})_{\text{exp}}$ is the experimental TOC decay (mg L$^{-1}$), $4.32 \times 10^7$ is a conversion factor for homogenization of units, m is the number of carbon atoms of CYT, I is applied current (A), t is the electrolysis time (h). n is the number of electrons consumed per CYT molecule during mineralization and was taken as 30 with the assumption of transformation of the N atoms in CYT molecule into $NH_4^+$ according to the mineralization reaction (Equation (15)).

$$C_9H_{13}N_3O_5 + 13H_2O \rightarrow 9CO_2 + 3NH_4^+ + 27H^+ + 30e^- \tag{15}$$

EC was calculated by using Equation (16) [23]:

$$\text{EC}\left(\text{kWh}(\text{g TOC})^{-1}\right) = \frac{E_{\text{cell}}\,I\,t}{V_S\,\Delta(\text{TOC})_{\text{exp}}} \tag{16}$$

where $E_{\text{cell}}$ is the mean potential difference of the cell (V).

The evolution of the short-chain carboxylic acids generated during electro-Fenton treatment as the final organic byproducts of the mineralization process were identified and quantified by the same HPLC equipped with a Biorad column (9 mm, 250 mm, 4.6 mm) coupled with a UV detector set to 220 nm. 4 mM $H_2SO_4$ was used as mobile phase in isocratic elution mode with a flow rate of 0.6 mL min$^{-1}$. For identification and quantification of carboxylic acids, a comparison of the retention time and peak area of standard solutions was performed.

Inorganic ions, $NO_3^-$ and $NH_4^+$, released in the solution during mineralization of CYT were identified and quantified by Dionex ICS-1000 (Thermo Scientific, Boston, USA) Basic Ion Chromatography System coupled with a Dionex DS6 conductivity detector containing a cell maintained at 35 °C. The data acquisition was carried out by Chromeleon software. The $NH_4^+$ content was detected and measured with a Dionex CS12A, 25 cm $\times$ 4 mm (i.d.) cationic column and a mobile phase of 30 mM methanesulfonic acid at 0.36 mL min$^{-1}$, whereas the $NO_3^-$ content was measured with a Dionex AS4A-SC, 25 cm $\times$ 4 mm (i.d.) anionic column and a mobile phase as a mixture of 1.8 mM $Na_2CO_3$ and 1.7 mM $NaHCO_3$ solution at 1.2 mL min$^{-1}$. The applied current in the SRS (self-regenerating suppressor) that was required to prevent the effect of the eluent ions in the detector signal was 30 mA in both cases.

In order to identify the cyclic/aromatic intermediates formed during the electro-Fenton treatment, 0.1 mM CYT solution was electrolyzed at 50 mA for 30 min. Samples for GC-MS were obtained by solvent extraction of organic components. Extraction of the resulting organics was performed three times with ethyl acetate followed by evaporation of the solution. Then, the organic fraction was again diluted with the same organic solvent. GC-MS analyses were performed using a GC-MS (Thermo Scientific, Boston, MA, USA) analyzer equipped with TRACE 1300 gas chromatography coupled to an ISQ single quadrupole mass spectrophotometer operating in electron impact mode at 70 eV. Samples were injected to the GC-MS using a TG-5Ms 0.25 mm, 30 m, 0.25 mm (i.d.) column. The temperature ramp applied for this column was 50 °C for 1 min, 10 °C min$^{-1}$ to 330 °C and kept at this

temperature for 5 min. The temperature of the inlet and detector were 200 and 250 °C, respectively. Helium was used as carrier gas at a flow rate of 1.2 mL min$^{-1}$.

## 4. Conclusions

According to the results obtained in the degradation kinetics studies, complete oxidative degradation of 0.1 mM (24.32 mg L$^{-1}$) CYT was achieved at 15 min under 300 mA at pH 3 using the electro-Fenton process with BDD/carbon felt electrode cells. Almost complete mineralization (97% of total organic carbon removal) of the same solution was obtained after 360 min under 500 mA constant current. The absolute rate constant of the oxidation reaction of CYT by hydroxyl radicals was determined using the competition kinetic method and found to be $5.35 \times 10^9$ M$^{-1}$ s$^{-1}$. A literature survey was carried out in order to compare the degradation/mineralization performance of the electro-Fenton process with previously published studies on the treatment of CYT. These data are summarized below in Table 3. As can be observed, the CYT degradation by electro-Fenton process seems to be more efficient than other strategies in terms of total degradation rate and mineralization efficiency.

**Table 3.** Degradation/mineralization performance of CYT with various methods.

| Method | Experimental Conditions | Removal (%) | Reference |
|---|---|---|---|
| UV/H$_2$O$_2$ | [CYT]$_0$ = 10 mg/L<br>[H$_2$O$_2$] = 400 μM<br>pH = 7 | 90% degradation in 120 min | [65] |
| UV/K$_2$S$_2$O$_8$ | [CYT]$_0$ = 10 mg/L<br>[K$_2$S$_2$O$_8$] = 200 μM<br>pH = 7 | 98% degradation in 60 min | [65] |
| Gamma radiation | [CYT]$_0$ = 10 mg/L<br>Dose rate = 1.66 Gy min$^{-1}$<br>pH = 7 | 95% degradation in 240 min | [66] |
| UV/TiO$_2$ + activated carbon | [CYT]$_0$ = 50 mg/L<br>Carbon mass = 5 mg<br>TiO$_2$ mass = 5 mg<br>pH = 7 | 90% degradation in 10 min | [67] |
| Photodegradation | [CYT]$_0$ = 10 mg/L<br>[K$_2$S$_2$O$_8$] = 100 μM<br>pH = 7 | 90% degradation in 30 min<br>45% mineralization in 120 min | [68] |
| Simulated solar photocatalysis | [CYT]$_0$ = 30 mg/L<br>[TiO$_2$] = 300 mg/L<br>[H$_2$O$_2$] = 90 mg/L<br>pH = 6.5 | 100% degradation in 30 min<br>80% mineralization in 360 min | [7] |
| Photo-Fenton-like treatment | [CYT]$_0$ = 30 mg/L<br>[Fe$^{3+}$] = 3 mg/L<br>[C$_2$K$_2$O$_4$·H$_2$O] = 90 mg/L<br>pH = 3 | 100% degradation in 30 min<br>82% mineralization in 360 min | [6] |
| Anodic oxidation | [CYT]$_0$ = 20 mg L$^{-1}$<br>Graphite anode<br>Current density = 10 mA cm$^{-2}$<br>[NaCl] = 50 mM<br>pH = 3 | 98% degradation in 30 min<br>60% mineralization in 180 min | [3] |
| Electro-Fenton (present work) | [CYT]$_0$ = 24.32 mgL$^{-1}$<br>BDD anode<br>Carbon felt cathode<br>[Na$_2$SO$_4$] = 50 mM<br>pH = 3 | 100% degradation in 15 min<br>91% mineralization in 240 min at 300 mA (9.375 mA cm$^{-2}$)<br>85% mineralization in 120 min at 500 mA (15.625 mA cm$^{-2}$)<br>95% mineralization in 240 min at 500 mA (15.625 mA cm$^{-2}$) | This study |



**Author Contributions:** S.C.: conceptualization: investigation, methodology, draft preparation, writing, formal analysis; B.Ö.: investigation: validation, writing; N.O.: supervision: methodology, formal analysis; C.T.: supervision: validation; M.A.O.: supervision: review and editing, project administration. All authors have read and agreed to the published version of the manuscript.

**Funding:** This research received no external funding.

**Data Availability Statement:** Date is contained within the article.

**Acknowledgments:** Sule Camcioglu and Baran Özyurt acknowledge TUBITAK (The Scientific and Technological Research Council of Turkey) for providing financial support through 2219-International Postdoctoral Research Fellowship Program for Turkish Citizens. Sule Camcioglu and Baran Ozyurt thank both Ankara University and Gustave Eiffel University for their support of their postdoctoral research.

**Conflicts of Interest:** The authors declare no conflict of interest.

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
