# Peer review of "Fast and Complete Destruction of the Anti-Cancer Drug Cytarabine from Water by Electrocatalytic Oxidation Using Electro-Fenton Process"

_catalysts, doi:10.3390/catal12121598_

Round 1
Reviewer 1 Report
The manuscript deserves publication, after minor revisions:
1) AOPs abbreviation in the keywords should be eliminated.
2) Eq. 1 repeats the number identification.
3) Eqs. are not homogeneously elaborated. Font is different, size, etc.
4) the identifications of Figures such as (a) (b) (c) and so on were anchored in wrong way in the figures. please, check it.
5) X-axis in the Figure 3 could be replaced by CYT concentration removal, TOC removal or other parameter, which could be different than the time, because no linear trend behavior will be observed, and better data analysis will be attained.
6) Sometimes the X-axis is reported in h or min, please, check it.
7) Fig 6 is not clear, and it could be improved by using a photo adapted a design system.
8) Table 1, column 'Method" should describe the method used in the present work. Should it be electro-Fenton?
Author Response
1) AOPs abbreviation in the keywords should be eliminated.
Response: This abbreviation is maintained, since it avoids a too long keyword (Electrochemical Advanced Oxidation Processes)
2) Eq. 1 repeats the number identification.
Response: Equation numbers on ‘Analytical procedures’ subsection (page 18) have been checked and corrected.
3) Eqs. are not homogeneously elaborated. Font is different, size, etc.
Response: Equations have checked and corrected consistently.
4) the identifications of Figures such as (a) (b) (c) and so on were anchored in wrong way in the figures. please, check it.
Response: The identification of figures such as (a), (b), (c) was checked and corrected.
5) X-axis in the Figure 3 could be replaced by CYT concentration removal, TOC removal or other parameter, which could be different than the time, because no linear trend behavior will be observed, and better data analysis will be attained.
Response: We think the reviewer ask for the Fig. 2. Since TOC is presented as a normalized value, we have kept the presentation format but introduced the term "removal" in the legend.
6) Sometimes the X-axis is reported in h or min, please, check it.
Response: X axis is reported in h for mineralization experiments (which need long electrolysis times). On the contrary, X axis is reported in min for degradation kinetic experiments requiring short electrolysis times.
7) Fig 6 is not clear, and it could be improved by using a photo adapted a design system.
Response: Fig. 6 have been improved as suggested by the reviewer
8) Table 1, column 'Method" should describe the method used in the present work. Should it be electro-Fenton?
Response: We think this comment is related to the Table 3. It is appropriately modified as suggested by the Reviewer.
Reviewer 2 Report
Authors have demonstrated electrocatalytic oxidation of anti-cancer drug Cytara-2 bine from water using of electro-Fen-3 ton process, seems interesting. However, Authors need to address the following comments before to be published.
What are the rationale authors chosen the electrodes of BDD anode and Carbon felt cathode electrode?
What is the effect of temperature in the catalytic process?
Detailed information about the electrolyte and applied current in the experimental section is required. What happen when you change the electrolyte with different pH?
Have authors performed any real samples measurements?
Author Response
1) What is the effect of temperature in the catalytic process?
Response: The activation energy of the Fenton reaction is low. Therefore, the effect of temperature is not significant. On the other hand, the room temperature favors the electro-Fenton process since higher temperatures can lead to the decrease of oxygen concentration in water which is crucial for generation of H2O2. This parameter is important since it constitutes the rate determining step. Therefore, all experiments were carried out in room temperature (20 °C).
2) Detailed information about the electrolyte and applied current in the experimental section is required. What happen when you change the electrolyte with different pH?
Response: Thanks for these remarks. The section 3.2 was modified in order to add this information
3) Have authors performed any real samples measurements?
Response: The experiments have been performed in synthetic solutions. We will carry out the treatment of real samples in a next work.